# Knowledge and practices regarding diabetic retinopathy among diabetic patients registered in a chronic disease management system in eastern China

**Fang Duan**[1☯], **Yan Zheng**[2☯], **Qian Zhao**[2], **Ze Huang**[2], **Yuedan Wu**[2], **Guoyi Zhou**[2]*, **Xiang Chen**[1]*

**1** Zhongshan Ophthalmic Center, State Key Laboratory of Ophthalmology, Sun Yat-sen University, Guangzhou, China, **2** Affiliated Yueqing Hospital, Wenzhou Medical University, Wenzhou, Zhejiang, China

☯ These authors contributed equally to this work.
* chen1094@hotmail.com (XC); zhougy123456@163.com (GZ)

**Data Availability Statement:** Relevant data that support the findings of this study are in the paper.

## Abstract

### Purpose

To investigate the knowledge and practices regarding diabetic retinopathy (DR) among diabetic patients included in a community-based primary health system (CBPHS) in China.

### Methods

Diabetic patients aged 18 years and above registered in the CBPHS in Yueqing city, Zhejiang province were recruited. Information obtained by questionnaire included: demographic and socioeconomic status, knowledge about DR, and ocular and medical history. The primary outcome was whether the participant knew that DM can affect the eyes, defined according to the question: "Do you know diabetes mellitus (DM) can affect eyes? (yes or no)". A knowledge score was calculated based on the responses to seven questions, with 1 point awarded for a correct response and 0 points for an incorrect or uncertain answer.

### Results

A total of 1972 diabetic patients were included in the study with an average age of 65.2±10.8 years, 45.7% were male. One thousand two hundred and nineteen patients (61.8%) knew that DM can affect the eyes. Significant differences in age, education, income status, insurance covering eye care, fasting blood glucose, duration of DM, history of hypertension existed between subjects who knew and those who did not know that DM can affect the eyes (P<0.05 for all). The proportion of correct answers to the DR knowledge questions ranged from 33.3% to 61.8%, with an average score of 3.65±2.47. In the multiple regression analysis, the knowledge score was significantly associated with age, education, income, history of hypertension, duration of DM, being told that regular examinations should be performed and concern about vision loss (P <0.01 for all).

The minimal dataset required to reproduce the results of the paper are not publicly available due to research participants did not consent to have their individual data publicly shared. The data are available on request from the Yueqing Hospital Ethical Review Board, please direct requests to the ethics committee coordinator Miss CHEN, qianyunchen@aliyun.com.

**Funding:** This study was supported partly by National Natural Science Foundation of China (81400381) and Zhejiang Medical and Health Science and Technology Project (2018269795).

**Competing interests:** The authors declare no competing interests.

## Conclusions

The knowledge toward DR among DM patients were still low within the chronic disease management system in eastern China. Routine ophthalmic screening, health care promotions, and educational programs should be emphasized and implemented for better DR prevention and management.

## Introduction

Diabetes mellitus (DM) is a major public health problem worldwide, and its prevalence is increasing at an alarming rate with population growth and aging [1]. As reported by the International Diabetes Federation (IDF), there were 451 million diabetic patients worldwide in 2017, which is expected to increase to 693 million by 2045 [2]. It was reported that the prevalence of DM in China reached 10.9% in 2013, which was nearly 10-fold of that in the 1980s, and that for prediabetes was 35.7% [3]. Diabetic retinopathy (DR) is a leading cause of vision loss worldwide and a common ocular complication of DM, occurring in one of three diabetic patients [4–7]. However, previous studies have found that nearly half of the DR patients in China have never received any ocular examination [8].

Early detection and intervention of DR had been showed to be critical to prevent irreversible blindness and improve the patient's quality of life [9]. And the efficacy and cost effectiveness of early detection and treatment of DR had been well established [10]. Previous studies found that a lack of DR knowledge was associated with poor patients adherence [8, 11]. In our previous study, we used an in-depth interview method to explore the influencing factors on the compliance of timely visits among patients with proliferative diabetic retinopathy, and found that more than 90% of the patients were lacking knowledge about DR [12]. A recent Cochrane review showed that interventions to increase awareness about DR are vital in improving attendance for DR screening, and thus a potentially important solution for reducing blindness caused by DR [13]. Therefore, patient awareness and knowledge about DR will be the key to successful disease management and prevention.

In order to support patients with chronic diseases, a community-based primary health system (CBPHS) was introduced throughout the entire nation to enable access to basic and less costly healthcare services, especially after the New Health Reform in 2009, which called for medical insurance coverage for more than 90% of Chinese people [14]. Hospital-based model has been shifted to delivery in primary settings. The CBPHS has proven to be very helpful to manage chronic diseases [15]. Free regular blood glucose tests are provided for diabetes patients registered in the CBPHS, but DM complications are not necessarily assessed. Since the previous studies on the patient awareness and knowledge of DR were based on general population, and the CBPHS has been carried out for several years in China, it is necessary to evaluate the potential changes CBSPHS may bring in the DM patients. Therefore, the present study aimed to evaluate the awareness and knowledge about DR, as well as the associated risk factors, among DM populations registered in the CBPHS in China. The information will be very useful to better inform future policy-making regarding DR prevention and treatment.

## Methods

Yueqing City is located in the east of Zhejiang Province, China, covering an area of 1286.90 square kilometers with a population of over 1 million. Yuecheng community and Nanyue

community were selected to represent the urban and suburban areas of Yueqing City, respectively. Patients aged 18 years and above with a history of physician-diagnosed DM who were registered in the CBPHS were invited to participate in this study. In detail, there were 3156 DM patients in Yuecheng community and 1300 participants were selected by simple random sampling method. There were 923 DM patients in Nanyue community and all of them were invited to participate. The exclusion criteria included a DM history of less than 12 months and inability to cooperate with the interviewer. Questionnaires were administered by a trained community physician. The study was conducted between May and September 2017. Ethical approval was obtained from the Yueqing Hospital Ethical Review Board, and the study adhered to the Declaration of Helsinki. Written informed consent was received from all participants.

The study questionnaire was modified according to a previous study [8], consisting of three parts. The first part included questions assessing the patient's demographic and socioeconomic status, including age, gender, contact information, education level, monthly income, and information of medical insurance. The second part assessed the patient's medical history of DM, including the type, duration, diagnosis, and history of DM-related diseases. The third part assessed the patient's knowledge and actual practice regarding DR, including a knowledge section (7 questions about DR knowledge, e.g., Does DM affect eyes? How to know DM has affected your eyes?).

The primary outcome of the study was whether the participant knew that DM can affect the eyes, defined according to the question "Do you know diabetes mellitus can affect eyes? (yes or no)". Categorical and continuous outcomes were compared using Chi-squared tests and $t$ tests, respectively. To assess the participants' knowledge about DR, a knowledge score was calculated based on the responses to 7 questions in the knowledge section, with 1 point awarded for a correct response and 0 points for an incorrect or uncertain answer. All variables significant at the 0.05 level in univariate analysis then were entered into multiple regression. Multiple linear regression was performed to estimate the association between the knowledge score and other factors, such as the demographic and socioeconomic status and DM history. A P value <0.05 was considered to be statistically significant. All statistical analyses were performed using Stata 12.0 (StataCorp, College Station, TX, USA).

## Results

A total of 1972 diabetic patients completed the questionnaires in the two communities, 1151 from Yuecheng community and 821 from Nanyue community. The mean (standard deviation) age was 65.2±10.8 years, and 45.7% were male. Among all study participants, 1219 patients (61.8%) knew that DM can affect the eyes. As shown in Table 1, subjects who knew that DM can affect the eyes were significantly younger than those who did not know (P<0.001). There were significant differences in education, income status and the insurance covering eye care between subjects who knew and those who did not know that DM can affect the eyes (P<0.001, P = 0.028 and P<0.001, respectively). No significant difference existed regarding the insurance covering DM between patients in these two groups.

As shown in Table 2, most patients in both groups had type 2 DM (96.9%). Subjects who knew that DM can affect the eyes had a higher fasting blood glucose and longer duration of diabetes compared to those who did not know (both P < 0.001). Approximately half of the subjects knew that they had DM through their physical check-up, 48.8% through presenting discomfort in the body and less than 1% through presenting discomfort in the eyes. Subjects who did not know that DM can affect the eyes had a lower proportion of using insulin and more concerns with using insulin (P<0.001 and P = 0.002, respectively). In addition, more

**Table 1. Demographic and socioeconomic status of study participants.**

| Factors | All (n = 1972) | Did not know DM affects eyes (n = 753) | Knew DM affects eyes (n = 1219) | P* |
|---|---|---|---|---|
| **Age (years), mean (SD)** | 65.2 (10.8) | 66.8 (10.7) | 64.1 (10.7) | <0.001† |
| **Males, n (%)** | 902 (45.7) | 309 (41.1) | 593 (48.7) | 0.001‡ |
| **Education, n (%)** | | | | <0.001§ |
| None | 541 (27.4) | 259 (34.4) | 282 (23.1) | |
| Elementary school | 641 (32.5) | 241 (32.0) | 400 (32.8) | |
| Junior high school | 447 (22.7) | 150 (19.9) | 297 (24.4) | |
| High school | 273 (13.8) | 75 (9.96) | 198 (16.2) | |
| College or above | 70 (3.55) | 28 (3.72) | 42 (3.45) | |
| **Monthly family income (USD), n (%)** | | | | 0.028§ |
| 1–300 | 227 (11.5) | 112 (14.9) | 115 (9.43) | |
| 301–450 | 327 (16.6) | 124 (16.5) | 203 (16.7) | |
| 451–750 | 475 (24.1) | 156 (20.7) | 319 (26.2) | |
| 751–1500 | 648 (32.9) | 269 (35.7) | 379 (31.1) | |
| >1500 | 295 (15.0) | 92 (12.2) | 203 (16.7) | |
| **Insurance covering DM, n (%)** | | | | 0.860‡ |
| Self-pay | 87 (4.41) | 34 (4.52) | 53 (4.35) | |
| Social health insurance | 1,885 (95.6) | 719 (95.5) | 1,166 (95.7) | |
| **Insurance covering eye care, n (%)** | | | | <0.001‡ |
| Yes | 423 (21.5) | 156 (20.7) | 267 (21.9) | |
| No | 623 (31.6) | 204 (27.1) | 419 (34.4) | |
| Not sure | 926 (47.0) | 393 (52.2) | 533 (43.7) | |

SD: standard deviation, 1 USD = 6.67 RMB, DM: diabetes mellitus

* P values were for comparing participant characteristics between the two groups.

† t test was used.

‡ Chi-squared test was used.

§ Ordinal logistic regression was used.

subjects who knew that DM can affect the eyes had a history of hypertension (P = 0.012), while the history of hypercholesterolemia was similar between the two groups.

The prevalence of diagnosed DR was significantly higher among subjects who knew that DM can affect the eyes (P<0.001) (Table 3). Most subjects who knew that DM can affect the eyes obtained instructions for regular eye examinations from physicians (57.8%). There were 1587 patients (80.9%) who had never undergone an eye examination, and only 178 patients (9.07%) underwent a yearly eye examination. The frequency of eye examinations was significantly different between the two groups (P<0.001). The main obstacle to having an eye examination was "vision is not affected" for all subjects, accounting for 41.1%. A lack of company was another important barrier to having an eye examination for the study participants in both groups. Regarding how the subjects knew that DM had affected their eye, subjects who knew through a fundus examination after pupil dilation only accounted for 5.98%; most subjects knew through discomfort in their eyes (43.1%). Only 29.5% of subjects thought that DM patients should have their eyes examined annually, and this percentage was significantly higher among subjects who knew that DM can affect the eyes. Additionally, 8.77% of subjects thought there was no need to have their eyes checked because of diabetes, and this percentage was significantly higher among subjects who did not know that DM can affect the eyes. Furthermore, people who did not know that DM can affect the eyes tended to be less anxious toward losing vision (P<0.001).

**Table 2. Medical history of diabetes among study participants by their knowledge of diabetes.**

| | All (n = 1972) | Did not know DM affect eyes (n = 753) | Knew DM affect eyes (n = 1219) | P* |
|---|---|---|---|---|
| **DM type, n (%)** | | | | 0.051† |
| Type 1 | 61 (3.1) | 16 (2.1) | 45 (3.7) | |
| Type 2 | 1,911 (96.9) | 737 (97.9) | 1174 (96.3) | |
| **Fasting blood glucose (mmol/L) Mean (SD)** | 7.00 (1.45) | 6.86 (1.28) | 7.08 (1.54) | 0.001‡ |
| **Years since DM diagnosed** | 6.24 (4.38) | 5.52 (3.96) | 6.69 (4.57) | <0.001‡ |
| **How to know you have DM, n (%)** | | | | <0.001§ |
| Physical check-up | 999 (50.7) | 423 (56.2) | 576 (47.3) | |
| Discomfort in the body | 963 (48.8) | 326 (43.3) | 637 (52.3) | |
| Discomfort in the eyes | 10 (0.51) | 4 (0.53) | 6 (0.49) | |
| **Using insulin, n (%)** | 184 (9.33) | 46 (6.11) | 138 (11.3) | <0.001† |
| **Concerns with using insulin, n (%)** | | | | 0.002¶ |
| None | 146 (7.56) | 70 (9.50) | 76 (6.37) | |
| Barely | 525 (27.2) | 167 (22.7) | 358 (30.0) | |
| Some | 848 (43.9) | 303 (41.1) | 545 (45.7) | |
| Substantial | 411 (21.3) | 197 (26.7) | 214 (17.9) | |
| **History of hypertension, n (%)** | 1,175 (59.6) | 422 (56.0) | 753 (61.8) | 0.012† |
| **History of hypercholesterolemia, n (%)** | | | | 0.134† |
| No | 1,335 (67.7) | 519 (68.9) | 816 (66.9) | |
| Yes | 244 (12.4) | 79 (10.5) | 165 (13.5) | |
| Not sure | 393 (19.9) | 155 (20.6) | 238 (19.5) | |

SD: standard deviation

* P values were for comparing the participants' medical history between the two groups.

† Chi-squared test was used.

‡ *t* test was used.

§ Fisher's exact test was used.

¶ Ordinal logistic regression was used.

The subjects' knowledge about DR are shown in Table 4; the proportion of correct answers to the other 6 questions regarding DR knowledge (besides DM can affect the eyes) ranged from 33.3% to 61.0%, with an average score of 3.65±2.47. The association between the knowledge score and other risk factors is shown in Table 5. In the multiple regression analysis, the knowledge score was significantly associated with age, education, income, history of hypertension, duration of DM, being told that regular examinations should be performed and concern about vision loss. In detail, age was negatively associated with knowledge score (β = -0.02, 95% CI: -0.03, -0.01, P<0.001). The higher level of education was associated with higher score of knowledge: elementary school vs. none, β = 0.54, 95% CI: 0.30, 0.78, P<0.001; junior high school vs. none, β = 0.97, 95% CI: 0.69, 1.24, P<0.001; high school vs. none, β = 0.96, 95% CI: 0.64, 1.29, P<0.001; college or above vs. none, β = 1.28, 95% CI: 0.76, 1.79, P<0.001. Higher level of family income was associated with higher score: 301–450 USD vs. < = 300USD, β = 0.55, 95%CI: 0.21, 0.89, P = 0.001; 451–750 USD vs. < = 300USD, β = 0.46, 95%CI: 0.14, 0.78, P = 0.005. History of hypertension and years since DM diagnosed were positive associated knowledge score (β = 0.27, 95% CI: 0.09, 0.46, P = 0.003 and β = 0.04, 95% CI: 0.02, -0.06, P<0.001 respectively). Told regular eye examination should be done and concern about vision loss were positive associated knowledge score (β = 2.07, 95% CI: 1.87, 2.28, P<0.001 and β = 1.60, 95% CI: 1.37, 1.83, P<0.001 respectively).

**Table 3. History of ophthalmic care among diabetic patients by their knowledge of diabetes.**

| | All (n = 1972) | Did not know DM affects eyes (n = 753) | Knew DM affects eyes (n = 1219) | P* |
|---|---|---|---|---|
| **Diagnosed with DR, n (%)** | 95 (4.82) | 14 (1.86) | 81 (6.64) | <0.001† |
| **Told regular eye examination should be done, n (%)** | | | | <0.001† |
| Never | 613 (31.1) | 434 (57.6) | 179 (14.7) | |
| Yes, only by physician | 966 (49.0) | 261 (34.7) | 705 (57.8) | |
| Yes, only by ophthalmologist | 95 (4.82) | 9 (1.20) | 86 (7.05) | |
| Yes, by physician and ophthalmologist | 298 (15.1) | 49 (6.51) | 249 (20.4) | |
| **Frequency of eye examinations, n (%)** | | | | <0.001‡ |
| Never | 1,587 (80.9) | 673 (90.2) | 914 (75.2) | |
| More than 5 years | 56 (2.85) | 18 (2.41) | 38 (3.13) | |
| Every 3 to 5 years | 61 (3.11) | 12 (1.61) | 49 (4.03) | |
| Every 2 years | 57 (2.91) | 13 (1.74) | 44 (3.62) | |
| Yearly | 178 (9.07) | 28 (3.75) | 150 (12.3) | |
| More often than yearly | 23 (1.17) | 2 (0.27) | 21 (1.73) | |
| **Told the time of next follow-up visit, among participants who ever had eye examination, n (%)** | 246 (63.9) | 29 (36.3) | 217 (71.2) | <0.001† |
| **Obstacle to having an eye examination, n (%)** | | | | <0.001§ |
| No time | 380 (19.3) | 110 (14.6) | 270 (22.2) | |
| No company | 603 (30.6) | 209 (27.8) | 394 (32.3) | |
| Poor transportation | 96 (4.87) | 32 (4.25) | 64 (5.25) | |
| Unreliable doctor | 40 (2.03) | 23 (3.05) | 17 (1.39) | |
| No money | 42 (2.13) | 6 (0.80) | 36 (2.95) | |
| Vision is not affected | 811 (41.1) | 373 (49.5) | 438 (35.9) | |
| **How to know DM had affected eyes, n (%)** | | | | <0.001† |
| Vision test | 450 (22.8) | 177 (23.5) | 273 (22.4) | |
| Vision and ocular surface | 139 (7.05) | 43 (5.71) | 96 (7.88) | |
| Discomfort in the eye, such as pain and blurry vision | 850 (43.1) | 280 (37.2) | 570 (46.8) | |
| Fundus examination after pupil dilation | 118 (5.98) | 8 (1.06) | 110 (9.02) | |
| Not sure | 415 (21.0) | 245 (32.5) | 170 (14.0) | |
| **How often should diabetics have their eyes examined, n (%)** | | | | <0.001† |
| Yearly | 581 (29.5) | 141 (18.7) | 440 (36.1) | |
| Every 2 years | 318 (16.1) | 84 (11.2) | 234 (19.2) | |
| 3–5 years | 91 (4.61) | 36 (4.78) | 55 (4.51) | |
| More than every 5 years | 96 (4.87) | 24 (3.19) | 72 (5.91) | |
| Never | 173 (8.77) | 110 (14.6) | 63 (5.17) | |
| Not sure | 713 (36.2) | 358 (47.5) | 355 (29.1) | |
| **Concern about vision loss, n (%)** | | | | <0.001§ |
| Never | 433 (22.0) | 328 (43.6) | 105 (8.61) | |
| Rarely | 803 (40.7) | 284 (37.7) | 519 (42.6) | |
| Sometimes | 562 (28.5) | 117 (15.5) | 445 (36.5) | |
| Often | 130 (6.59) | 19 (2.52) | 111 (9.11) | |
| Very often | 44 (2.23) | 5 (0.66) | 39 (3.20) | |

* P values were for comparing the history of ophthalmic care between the two groups.

† Chi-squared test was used.

‡ Ordinal logistic regression was used.

§ Fisher's exact test was used.

**Table 4. Knowledge about diabetic retinopathy among diabetic patients (n = 1972).**

| | Correct | Correct proportion (%) |
|---|---|---|
| **Knowledge questions, n (%)** | | |
| DM can affect eyes | 1219 | 61.8 |
| DM can cause blindness | 999 | 50.7 |
| DR is preventable | 1145 | 58.1 |
| DR is treatable | 1202 | 61.0 |
| Diabetic patients are more likely to get eye disease | 1105 | 56.0 |
| DR usually has early symptoms | 659 | 33.4 |
| Regular eye examinations are necessary | 866 | 43.9 |

**Table 5. Linear regression model for risk factors of participant's knowledge score about diabetic eye diseases (n = 1972).**

| | Simple linear regression | | Multiple linear regression | |
|---|---|---|---|---|
| | Beta coefficient (95% CI) | *P* | Beta coefficient (95% CI) | *P* |
| **Age (years)** | -0.03 (-0.04, 0.02) | <0.001 | **-0.02 (-0.03, -0.01)** | **<0.001** |
| **Males** | 0.32 (0.11, 0.54) | 0.004 | -0.001 (-0.19, 0.19) | 0.991 |
| **Education** | | | | |
| None | Reference | | Reference | |
| Elementary school | 0.51 (0.23, 0.79) | <0.001 | **0.54 (0.30, 0.78)** | **<0.001** |
| Junior high school | 0.90 (0.59, 1.20) | <0.001 | **0.97 (0.69, 1.24)** | **<0.001** |
| High school | 1.08 (0.72, 1.43) | <0.001 | **0.96 (0.64, 1.29)** | **<0.001** |
| College or above | 1.18 (0.58, 1.79) | <0.001 | **1.28 (0.76, 1.79)** | **<0.001** |
| **Monthly family income (USD)** | | | | |
| 1–300 | Reference | | Reference | |
| 301–450 | 0.77 (0.35, 1.18) | <0.001 | 0.55 (0.21, 0.89) | 0.001 |
| 451–750 | 0.87 (0.48, 1.25) | <0.001 | 0.46 (0.14, 0.78) | 0.005 |
| 751–1500 | 0.44 (0.07, 0.81) | 0.021 | 0.08 (-0.25, 0.40) | 0.649 |
| >1500 | 1.10 (0.68, 1.52) | <0.001 | 0.25 (-0.11, 0.61) | 0.168 |
| **Yuecheng community (Nanyue community as reference)** | 0.19 (-0.03, 0.41) | 0.090 | | |
| **Type 2 DM (type 1 as reference)** | -0.55 (-1.18, 0.08) | 0.087 | | |
| **Fasting blood glucose (mmol/L)** | 0.05 (-0.03, 0.12) | 0.227 | | |
| **Using insulin (yes/no)** | 0.94 (0.57, 1.31) | <0.001 | 0.28 (-0.03, 0.58) | 0.078 |
| **History of hypertension (yes/no)** | 0.25 (0.02, 0.47) | 0.030 | **0.27 (0.09, 0.46)** | **0.003** |
| **Years since DM diagnosed** | 0.07 (0.04, 0.09) | <0.001 | **0.04 (0.02, 0.06)** | **<0.001** |
| **Diagnosed as DR (yes/no)** | 1.08 (0.57, 1.58) | <0.001 | 0.31 (-0.11, 0.73) | 0.149 |
| **Used to have eye examinations (yes/no)** | 0.96 (0.69, 1.23) | <0.001 | 0.14 (-0.10, 0.38) | 0.244 |
| **Told regular eye examinations should be done (never vs combined other options, never as reference)** | 2.75 (2.55, 2.95) | <0.001 | **2.07 (1.87, 2.28)** | **<0.001** |
| **Concern about vision loss (never vs combined other options, never as reference)** | 2.56 (2.32, 2.80) | <0.001 | **1.60 (1.37, 1.83)** | **<0.001** |

1 USD = 6.67 RMB, DM: diabetes mellitus, DR: diabetic retinopathy

[†] All variables with P<0.05 in the simple regression analysis were included in the multiple regression analysis.

## Discussion

In this study, we investigated the knowledge and practices regarding DR among adult DM patients registered in the CBPHS in eastern China via questionnaires, and found approximately 50% of the participants knew the knowledge about DR. Knowledge about DR was

related to age, education, family income, hypertension, diabetic duration, concern about vision loss and being told that regular eye examinations should be performed. Our results showed that despite the local residents were given free regular blood glucose tests and other types of DM management through the existing chronic disease management system, but more than one third of patient still knew little of DR and its prevention. This finding reveals how to improve knowledge of DR within the chronic disease management system is the key to battle with this preventable blindness-causing disease.

In the current study, all the study participants were recruited from the local CBPHS system. We found that only 61.8% of the participants knew that DM can affect the eyes, and only 50.7% knew that DM can cause blindness. The proportion of correct answers to the other questions ranged from 33.4% to 61%, indicating a poor to intermediate overall level of knowledge regarding DR among DM patients. This finding is similar to another population-based study in a suburban area (40.7%) [16], but lower than that reported in Shanghai (82.3%) [17]. However, there was no available previous data could be used to directly compare with CBPHS; therefore, our findings only provide some preliminary findings of the impact of CBPHS on DR knowledge. There are large variabilities in the reported knowledge level about DR in the literature: 37% in Australia, 65% in the USA and 27% in India [18–20]. A hospital-based study in Zhejiang and a community-based study in Liaoning, China reported that 67% and 68% of the study subjects, respectively, were unaware that diabetes can affect the eyes [21, 22]. The differences among these studies could be partly explained by differences in health care systems, study populations and methods. Despite numerous health education programs on diabetes having been implemented, the IDF report shows the most DM patients still lack sufficient knowledge about the complications associated with diabetes [23]. Studies have reported that the most common complication known by DM patients is heart disease, followed by cerebrovascular and renal disease, while fewer people are aware that DM can also affect the eyes [24]. In our study, people who had been diagnosed with DR were more likely to know that DM can affect the eyes than those who had not. However, given 80.9% patients never underwent an eye examination in our study, the diagnosed rate of DR (4.82%) may be far less than the real DR prevalence. One study in urban China found that more than 90% of subjects with DR were unaware that they had been affected by this eye condition [25], and the Beijing Eye Study reported that only 15% of the study participants were aware of their DR [26].

We found that participants with a longer DM duration and higher fasting blood glucose had better awareness that DM can affect the eyes. This finding is of no surprise, as both the duration of DM and blood glucose levels are known risk factors for DM-related complications [25, 27]. Furthermore, people with more severe DM and a longer duration of DM may have stronger initiatives and more opportunities to receive education regarding DM. Hypertension is another proven risk factor for DM and related complications [28], and people in our study with a history of hypertension also showed a better awareness that DM can affect the eyes. Even though dyslipidemia has been found to be a risk factor for DM [29], a history of hyper-cholesterolemia was not relevant to DR knowledge in our study. Patients with a better socio-economic background, including higher education level and income, had significantly better knowledge regarding DR compared to others. This is consistent with previous studies and could be due to better access to knowledge and a higher capability to understand [24, 30].

An annual dilated fundus examination is recommended for all patients with type 2 diabetes, but the reported awareness of the importance of routine check-ups is poor, even in developed countries [31]. In our study, only 29.5% of DM patients thought that they should have an annual eye examination, and the proportion was even lower for subjects who did not know that DM can affect the eyes (18.7% vs 36.1%). An awareness of regular examinations does not

always lead to action. Only 10.2% of the study participants actually had yearly or more frequent eye examinations, which is much lower than that reported in United States (63%) [32], Switzerland (71%) [33] and Jordan (76%) [34]. No national DR screening program has been implemented in China. According to our study, 31.9% of the DM patients had never been told that a regular eye examination is necessary, and 80.9% of the DM patients had never had their eyes examined. The major barriers for eye examinations include a perception that their vision is not affected, followed by a lack of time and lack of company, suggesting that more educational programs and health support are needed to increase the service uptake among DM patients. Therefore, education about DR knowledge and the importance of regular eye exams should be enhanced in the CBPHS.

Physicians play an important role in imparting awareness and knowledge about DR. Our study found that most people knew about the necessity of regular eye examinations from a physician, and secondly, from an ophthalmologist. Studies have also shown that physicians constitute the most important source of information in the knowledge gap for DM patients [35, 36]. In our study, only 6% of the participants knew that DM had affected their eye by a fundus examination after pupil dilation, and this proportion was as low as 1% for people who did not know that DM can affect the eyes. Most people knew that their eyes had been affected only after they experienced discomfort in the eye (43.1%), which could represent that the disease has evolved to later stages. Early detection and intervention are of vital importance to prevent sight-threatening and irreversible complications of DR. Thus, community-based and hospital-based educational programs, including posters, pamphlets, and screening camps, could be helpful for enhancing patients' awareness and knowledge to improve attitudes and practice. We believe that our study has highlighted the need for promoting knowledge regarding DR among DM patients, as well as shed light on potential barriers and interventions for policy-making in the future.

A limitation of this study concerns the representativeness of the study population, this study is based on the CBPHS and not strictly population-based. In addition, this study was cross-sectional in design, and further research is needed to investigate the short- and long-term health outcomes for DM patients with different levels of knowledge and practices toward DR.

In conclusion, with the existing chronic disease management system (free regular blood glucose tests and other types of DM management), the knowledge and practices regarding DR still haven't been improved. Routine ophthalmic screening and management of DR in DM patients should be emphasized, efforts should be directed toward health care promotions, and educational programs should be implemented for better patient outcomes. A more integrated and effective chronic disease management system is needed and should be improved gradually.

## Acknowledgments

We thank statistician Ling Jin, who provided the help for statistical analysis.

## Author Contributions

**Conceptualization:** Guoyi Zhou, Xiang Chen.

**Data curation:** Yan Zheng, Qian Zhao, Yuedan Wu.

**Formal analysis:** Fang Duan.

**Funding acquisition:** Fang Duan, Guoyi Zhou.

**Investigation:** Yan Zheng, Qian Zhao, Ze Huang.

**Methodology:** Fang Duan, Yan Zheng.

**Project administration:** Xiang Chen.

**Resources:** Guoyi Zhou.

**Supervision:** Xiang Chen.

**Validation:** Ze Huang, Yuedan Wu.

**Writing – original draft:** Fang Duan.

**Writing – review & editing:** Guoyi Zhou, Xiang Chen.

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
