## [Decision Letter · Decision Letter 0]

16 Apr 2020

PONE-D-19-29504

Knowledge and attitudes regarding diabetic retinopathy among diabetic patients registered in a chronic disease management system in eastern China

PLOS ONE

Dear Mr. Chen,

Thank you for submitting your manuscript to PLOS ONE. After careful consideration, we feel that it has merit but does not fully meet PLOS ONE’s publication criteria as it currently stands. Therefore, we invite you to submit a revised version of the manuscript that addresses the points raised during the review process.

We would appreciate receiving your revised manuscript by May 31 2020 11:59PM. To enhance the reproducibility of your results, we recommend that if applicable you deposit your laboratory protocols in protocols.io, where a protocol can be assigned its own identifier (DOI) such that it can be cited independently in the future. For instructions see: http://journals.plos.org/plosone/s/submission-guidelines#loc-laboratory-protocols

We look forward to receiving your revised manuscript.

Kind regards,

Wen-Jun Tu

Academic Editor

PLOS ONE

Journal Requirements:

"This study was supported partly by National Natural Science Foundation of China (81400381) and Zhejiang Medical and Health Science and Technology Project (2018269795)."

"The authors received no specific funding for this work."

Reviewers' comments:

Reviewer's Responses to Questions

**Comments to the Author**

1. Is the manuscript technically sound, and do the data support the conclusions?

Reviewer #1: Yes

Reviewer #2: Yes

Reviewer #3: Partly

2. Has the statistical analysis been performed appropriately and rigorously? 

Reviewer #1: No

Reviewer #2: Yes

Reviewer #3: No

3. Have the authors made all data underlying the findings in their manuscript fully available?

Reviewer #1: No

Reviewer #2: No

Reviewer #3: Yes

4. Is the manuscript presented in an intelligible fashion and written in standard English?

Reviewer #1: Yes

Reviewer #2: Yes

Reviewer #3: No

5. Review Comments to the Author

Reviewer #1: Dear authors,

I would like to congratulate the authors for their efforts in studying a topic that is so relevant. I would like to bring to their attention the following:

1. Sampling: The authors have mentioned that participants were randomly sampled. This has the potential to introduce bias, hence It would be beneficial to the readers if this was explained further.

2. Validation: Was the questionnaire validated? The questionnaire could be attached as a supplementary file.

3. Table 4: It is not clear what the authors are trying to depict in this table. The column “All” probably needs to be deleted.

4. One of the components of the knowledge score is “DM can cause blindness”. This should not be compared with “concern about vision loss” as both mean the same.

5. The same comment goes for “regular eye exams are necessary” which is a component of knowledge score, should not be compared with “told regular examinations should be done”

6. Strength of association: Although the authors comment on the significant association between knowledge score and various factors, there is however no comment on the strength of association. There appears to be only a week association between most of the factors except college education.

“concern about vision loss” and “told regular examinations should be done” also have a higher strength of association for obvious reasons as they are part of the knowledge score. These should not be compared.

Reviewer #2: Dear colleague. I enjoyed reading the manuscript. I have a few suggestions enclosed with the PDF as comments. I have a concern regarding use of the term "Knowledge and attitudes", when really, after reading the paper, i get a sense that knowledge, practices, risk factors for poor knowledge and lack of screening, and sources of information have been inquired. Also, I suggest adding an explanation about how variables were selected for the multiple linear regression.

Reviewer #3: The authors have used a questionnaire based approach to gather the information. They sought to find out the knowledge and attitudes of diabetic patients to the aspects of diabetic retinopathy. The authors have not scientifically adequately addressed the research question of attitudes. The study has not accounted for some confounding variables such as source of the information and access to information if formal or informal and if any person known to the patient or the person himself/herself already had retinopathy. The study does not add any scientifically sound useful information.

6. PLOS authors have the option to publish the peer review history of their article (what does this mean?). If published, this will include your full peer review and any attached files.

Reviewer #1: No

Reviewer #2: Yes: Dr. Vivek Gupta

Reviewer #3: No

---

## [Author Response · Author response to Decision Letter 0]

12 May 2020

Dear Editors and Reviewers:

Thank you for your valuable suggestions and comments concerning our manuscript entitled “Knowledge and practices regarding diabetic retinopathy among diabetic patients registered in a chronic disease management system in eastern China”. Those comments are all extremely helpful for revising and improving our paper. We have carefully made necessary corrections accordingly, and highlighted in the paper. 

Review Comments to the Author

  Reviewer #1: 

Dear authors, I would like to congratulate the authors for their efforts in studying a topic that is so relevant. I would like to bring to their attention the following:  1. Sampling: The authors have mentioned that participants were randomly sampled. This has the potential to introduce bias, hence It would be beneficial to the readers if this was explained further.

Response: Thanks for your suggestion. We totally agree that we need to maximize the validity of inferences from what was observed in the study sample to what is happening in the population. The participants in our study were selected by simple random sampling method. We had edited in our manuscript.

 2. Validation: Was the questionnaire validated? The questionnaire could be attached as a supplementary file.

Response: Thanks for asking this question. We actually did not validate that questionnaire since it has been used/validated in a previous published study from our institution (Ophthalmology.2010; 117:1755-62) with very few modifications. 

 3. Table 4: It is not clear what the authors are trying to depict in this table. The column “All” probably needs to be deleted.

Response: Thanks for your suggestion. We tried to display the responses of the participants to 7 different questions of DR knowledge. And we had deleted the column “All” in table 4 accordingly.

 4. One of the components of the knowledge score is “DM can cause blindness”. This should not be compared with “concern about vision loss” as both mean the same.

Response: Thanks for your comments. We were trying to answering this similar question from two different approaches here, the score of “DM can cause blindness” was hoping to understand more from knowledge level, “concern about vision loss” was hoping to understand it from psychological level, so that we can see whether knowledge or psychological effects will interact on the same question or not. 

 5. The same comment goes for “regular eye exams are necessary” which is a component of knowledge score, should not be compared with “told regular examinations should be done”

Response: Similar like comments No.4, we were trying to answering the same question from different approaches. “regular eye exams are necessary” was hoping to understand it from knowledge level, “told regular examinations should be done” was hoping to find out whether this knowledge had been given by health care professionals, since even it had been told, but people may still say we didn’t feel it is necessary. We just tried to understand it in details. 

 6. Strength of association: Although the authors comment on the significant association between knowledge score and various factors, there is however no comment on the strength of association. There appears to be only a week association between most of the factors except college education.

Response: Thanks for your great suggestion. We had added it in the results section.

 “concern about vision loss” and “told regular examinations should be done” also have a higher strength of association for obvious reasons as they are part of the knowledge score. These should not be compared.

Response: Please refer to our responses for comments No.4 and 5. 

  Reviewer #2: 

Dear colleague. I enjoyed reading the manuscript. I have a few suggestions enclosed with the PDF as comments. I have a concern regarding use of the term "Knowledge and attitudes", when really, after reading the paper, i get a sense that knowledge, practices, risk factors for poor knowledge and lack of screening, and sources of information have been inquired. Also, I suggest adding an explanation about how variables were selected for the multiple linear regression.

Response: Thanks for your kind words and encouragement. We had edited our manuscript according to your suggestions. We had changed the word “attitudes” to “practice” in the title. And we had added an explanation about how variables were selected for the multiple linear regression. For most of comments you made in PDF, we have revised it directly in the manuscript with highlights. 

I suggest that the authors avoid making a direct comparison with pre-CBPHS results of another region's population.

Response: Thanks for this valuable comments. We agree that a direct comparison may not be that appropriate, we have modified it accordingly. 

 Reviewer #3: 

The authors have used a questionnaire based approach to gather the information. They sought to find out the knowledge and attitudes of diabetic patients to the aspects of diabetic retinopathy. The authors have not scientifically adequately addressed the research question of attitudes. The study has not accounted for some confounding variables such as source of the information and access to information if formal or informal and if any person known to the patient or the person himself/herself already had retinopathy. The study does not add any scientifically sound useful information.

Response: Thanks for taking the time and efforts to review our manuscript, we appreciate your constructive comments and agree that we need to continue to improve the quality of this manuscript. Here are our responses to the comments:

We agree that the term “attitude” was not appropriate to use, it was the “knowledge” of those participants being studied mostly in the manuscript. Therefore the term of “practice” is now being used to replace “attitude”. 

Authors couldn’t agree anymore on the comments of potential confounding variables, those were indeed important factors we needed to address in this type of the observational study. The participants in our study were selected by simple random sampling method. Questionnaires were administered by a trained community physician, to help the participants fully understand the questions being asked. We have put some of potential confounding variables as the limitations in our manuscript (such as the source of the DM/DR information, the accessibility of the DM/DR information).

A few studies have been done to investigate the knowledge of DM/DR in general population in China, including a team from our institution, however, none was implemented after the established of chronic disease management system in 2009. The system was intended to manage patients and prevent complication, the aim of our paper was to access the current system’s influence on DM/DR knowledge fronts, and we hope our results can provide some new evidence on how to improve that chronic disease system. And we believe the cause of problem of DM/DR knowledge is quite universal. 

Again, we appreciate all of your insightful comments. Thank you for taking the time to help us improve the paper, really appreciated!

---

## [Decision Letter · Decision Letter 1]

2 Jun 2020

Knowledge and practices regarding diabetic retinopathy among diabetic patients registered in a chronic disease management system in eastern China

PONE-D-19-29504R1

Dear Dr. Chen,

We’re pleased to inform you that your manuscript has been judged scientifically suitable for publication and will be formally accepted for publication once it meets all outstanding technical requirements.

Kind regards,

Wen-Jun Tu

Academic Editor

PLOS ONE

Additional Editor Comments (optional):

Reviewers' comments:

Reviewer's Responses to Questions

**Comments to the Author**

1. If the authors have adequately addressed your comments raised in a previous round of review and you feel that this manuscript is now acceptable for publication, you may indicate that here to bypass the “Comments to the Author” section, enter your conflict of interest statement in the “Confidential to Editor” section, and submit your "Accept" recommendation.

Reviewer #2: All comments have been addressed

Reviewer #3: All comments have been addressed

2. Is the manuscript technically sound, and do the data support the conclusions?

Reviewer #2: Yes

Reviewer #3: Yes

3. Has the statistical analysis been performed appropriately and rigorously? 

Reviewer #2: Yes

Reviewer #3: Yes

4. Have the authors made all data underlying the findings in their manuscript fully available?

Reviewer #2: No

Reviewer #3: Yes

5. Is the manuscript presented in an intelligible fashion and written in standard English?

Reviewer #2: Yes

Reviewer #3: Yes

6. Review Comments to the Author

Reviewer #2: Thank You for addressing the concerns with the initial submission, that were raised in the first review.

Reviewer #3: The manuscript is better now. The authors have revised the paper keeping in view the reviewer comments.

7. PLOS authors have the option to publish the peer review history of their article (what does this mean?). If published, this will include your full peer review and any attached files.

Reviewer #2: Yes: Dr Vivek Gupta

Reviewer #3: No

---

## [Editor Report · Acceptance letter]

8 Jun 2020

PONE-D-19-29504R1 

Knowledge and practices regarding diabetic retinopathy among diabetic patients registered in a chronic disease management system in eastern China 

Dear Dr. Chen:

I'm pleased to inform you that your manuscript has been deemed suitable for publication in PLOS ONE. Congratulations! Your manuscript is now with our production department. 

Kind regards, 

on behalf of

Dr. Wen-Jun Tu 

Academic Editor

PLOS ONE